# More than Just Wine: The Nutritional Benefits of Grapevine Leaves

**DOI:** 10.3390/foods10102251

**Published:** 2021-09-23

**Authors:** Marisa Maia, Ana Rita Cavaco, Gonçalo Laureano, Jorge Cunha, José Eiras-Dias, Ana Rita Matos, Bernardo Duarte, Andreia Figueiredo

**Affiliations:** 1Grapevine Pathogen Systems Lab. (GPS Lab.), Biosystems and Integrative Sciences Institute (BioISI), Faculdade de Ciências, Universidade de Lisboa, Campo Grande, 1749-016 Lisboa, Portugal; mrmaia@fc.ul.pt (M.M.); arcavaco@fc.ul.pt (A.R.C.); gmlaureano@fc.ul.pt (G.L.); 2Departamento de Biologia Vegetal, Faculdade de Ciências da Universidade de Lisboa, Campo Grande, 1749-016 Lisboa, Portugal; armatos@fc.ul.pt (A.R.M.); baduarte@fc.ul.pt (B.D.); 3Estação Vitivinícola Nacional, Instituto Nacional de Investigação Agrária e Veterinária (INIAV), Quinta da Almoinha, 2565-191 Dois Portos, Portugal; jorge.cunha@iniav.pt (J.C.); eiras.dias@iniav.pt (J.E.-D.); 4MARE—Marine and Environmental Sciences Centre, Faculdade de Ciências da Universidade de Lisboa, Campo Grande, 1749-016 Lisboa, Portugal

**Keywords:** *Vitis vinifera* L., fatty acid content, elemental profile, pigments, nutrition

## Abstract

The domesticated species *Vitis vinifera* L. harbours many cultivars throughout the world that present distinctive phenology and grape quality. Not only have the grapes been used for human consumption, but the leaves are also used as a source of bioactive compounds and are present in the diets of several Mediterranean countries. We have selected seven different cultivars and performed elemental, fatty acid (FA) and pigment profiling. Total reflection X-ray fluorescence (TXRF) enabled the identification of 21 elements. Among them, Na, Ca and K were highly represented in all the cultivars and Zn was prevalent in *V. vinifera* cv. ‘Pinot noir’ and ‘Cabernet sauvignon’. Through gas chromatography, six FAs were identified, including omega-3 and omega-6 FA, the most abundant mainly in *V. vinifera* cv. ‘Tinta barroca’, ‘Pinot noir’ and ‘Cabernet sauvignon’. FA composition was used to determine nutritional quality parameters, namely atherogenic, thrombogenic, hypocholesterolemic/hypercholesterolemic and peroxidisability indexes as well as oxidability and oxidative susceptibility. Grapevine leaves were highlighted as a suitable source of health-promoting lipids. Given the popularity of “green” diets, we have also performed a leaf pigment analysis. Seventeen pigments including chlorophylls, *trans*-lutein, b-carotene and zeaxanthins were detected. ‘C19’ presented the highest content of most of the detected pigments.

## 1. Introduction

Linked to human culture since ancient times, the domesticated grapevine species (*Vitis vinifera* L.) became one of the most important and cultivated fruit crops in the world. It comprises up to 5000 cultivars which are mainly used for wine production and table grape commercialisation. This crop plays a key role in many countries’ economies, with a global market size of over EUR 30 billion [1].

While wine and grapes are the most known and valuable grapevine products, several waste products derive from this industry, such as must, grape water and leaves, the last being one of the most abundant waste products in the wine industry [2,3]. This sub-product is considered a delicacy in many countries, including in the Mediterranean Basin. In fact, Mediterranean countries such as Turkey, Greece and Bulgaria cultivate specific grapevine cultivars especially for fresh and preserved leaf consumption [4,5,6].

The Mediterranean-type diet has been demonstrated to have numerous health benefits, namely on cardiometabolic diseases, diabetes and preventing certain types of cancer [7]. Moreover, recently its impact in decreasing the risk of mental disorders, including depression, has also been highlighted [7]. A wider inclusion of this disregarded by-product in the human diet or its use as a source of bioactive compounds is a good strategy, not only to introduce an added value to a waste product but also to come upon the European Union and United Nations’ demands towards more sustainable agricultural approaches and circular economy (Goals of the 2030 Agenda for Sustainable Development) [8].

The increasing search for healthier diets as well as new products with health promoting characteristics, has drawn attention to grapevine leaves in the last few years. Grapevine leaves have already shown to be an excellent source of bioactive molecules, mainly phenolic compounds [9,10,11,12], and their antioxidant properties have been described to protect from and retard oxidative processes [12,13,14,15,16,17,18,19,20,21,22].

For instance, Pari and Suresh demonstrated that grapevine leaf extracts exert antioxidant protective effects, decreasing lipid peroxidation, in liver and kidney alcohol-induced oxidative damage in rats [14]. Moreover, grapevine leaf extracts were shown to reduce lipid and protein damages, induced by hydrogen peroxide, in the brains of rats [23]. Leaf consumption is also positively correlated with the control of inflammatory disorders, pain, bleeding and high blood pressure [18,23,24,25,26,27].

Due to increasing interest in the consumption of grapevine leaves, some studies have also focused on their culinary processes. Raw grapevine leaves are not edible; thus, they must undergo some culinary processes. This is quite challenging as it is essential to preserve their nutritional properties. In a recent study by Lima and co-workers, the leaves of *V. vinifera* cv. Malvasia Fina and ‘Touriga Franca’ were studied for their colour, pigments and volatile fraction changes after blanching and boiling for 60, 75 and 90 min [18,28]. The authors demonstrated that boiling the leaves for 60 min is considered the most adequate time for not losing their properties [18,28].

Taking all this information into account, it is important to value grapevine leaves for their use in food, pharmaceutical and cosmetic industries, or as functional food ingredients.

We have previously shown that leaves from the elite cultivar Pinot noir may be used as an important source of nutraceutical compounds, particularly antioxidant compounds. ‘Pinot noir’ leaves also showed to be a suitable source of fatty acids (FAs) in comparison to other dietary items traditionally considered good sources of essential FAs [29]. In the present study, we have further evaluated seven grapevine cultivars (‘Trincadeira’, ‘Cabernet sauvignon’, ‘Pinot noir’, ‘Tinta barroca’, ‘Castelão’, ‘Bastardo’ and ‘C19’) considering total FA composition, elemental profiling, pigments and exploited different nutritional quality indexes. These cultivars were selected based on their importance in winemaking and marketing both internationally as well as for the Portuguese market. *Vitis vinifera* cv. Cabernet sauvignon and ‘Pinot noir’ are in the TOP 15 of the most cultivated grapevines in the world, with over 340,000 and 110,000 ha of cultivated area, respectively [30]. The remaining cultivars are important in the Portuguese wine industry or are currently under study to be introduced in the market.

Our results demonstrate that, overall, grapevine leaves of the cultivars analysed have a high potential for human consumption as well as to be considered as sources of bioactive compounds.

## 2. Materials and Methods

### 2.1. Plant Material

Seven *Vitis vinifera* cultivars (‘Trincadeira’, ‘Cabernet sauvignon’, ‘Pinot noir’, ‘Tinta barroca’, ‘Castelão’, ‘Bastardo’ and ‘C19’) were used in this study (Table 1).

The origin, berry colour and type of accession associated with the *Vitis* genotypes were accessed through bibliographic searches following the classification of the International Organisation of Vine and Wine (https://www.oiv.int (accessed on 31 July 2021)), *Vitis* International Variety Catalogue VIVC (https://www.vivc.de/ (accessed on 31 July 2021)) and the information in the archive of the Portuguese Ampelographic Grapevine Collection (CAN, from the Portuguese Coleção Ampelográfica Nacional).

Plant material was harvested at the CAN (international code PRT051, established in 1988), at INIAV- Estação Vitivinícola Nacional (Dois Portos). CAN occupies nearly 2 ha of area with homogeneous modern alluvial soils (lowlands) as well-drained soil. For all accessions in the field, a unique cultivar rootstock was used—Selection Oppenheim 4 (SO4)—and each accession comes from one unique plant. Plants have the same trailing system (bilateral cordon, Royat), canopy maintenance and agricultural management.

For plant material collection, the best possible health status was guaranteed. No agrochemical compounds application was conducted one month before leaf harvesting. Young leaves were harvested from fully developed plants, immediately frozen in liquid nitrogen and stored at −80 °C until analysis. Five biological replicates were considered for analysis.

### 2.2. Fatty Acid Quantification and Calculation of Health Lipid Indices

Fatty acid methyl esters (FAME) were prepared by direct *trans*-methylation of FA using 50 mg ground leaves with the addition of 3 mL of methanol-sulfuric acid solution (39:1 *v*/*v*). The methylation reaction occurred at 70 °C for 1 h and was stopped by cooling. Methyl esters were recovered by adding 3 mL of petroleum ether and 2 mL of ultra-pure water, according to [33]. The organic phase was collected and dried at 37 °C under nitrogen atmosphere and resuspended in hexane. A total of 1 μL of sample was injected for each analysis. Quantitative analysis of FAME was achieved by gas chromatography (3900 Gas Chromatograph, Varian, Palo Alto, CA, USA) equipped with a hydrogen flame ionisation detector, using a fused silica capillary column (0.25 mm i.d. × 50 m, WCOT Fused Silica, CP-Sil 88 for FAME, Varian) at 210 °C, according to previously optimised conditions [34].

To access the nutritional quality of the lipids present in the leaves of the analysed grapevine genotypes, the atherogenicity (AI) and thrombogenicity (TI) indexes, oxididability (Cox), oxidative susceptibility (OS), hypocholesterolemic/hypercholesterolemic index (h/H) and peroxidisability index (PI) were calculated according to the following equations [35,36,37,38,39]:(1)$$AI=\frac{\left(C16:0+C18:0\right)}{MUFAs+PUFAs_{n3}+PUFAs_{n6}}$$
(2)$$TI=\frac{\left(C16:0+C18:0\right)}{(0.5\times MUFAs+0.5\times PUFAs_{n6}+3\times PUFAs_{n3}+\left(\frac{PUFAs_{n3}}{PUFAs_{n6}}\right)}$$
(3)$$C_{ox}=\frac{\left(C18:1+10.3\times C18:2+21.6\times C18:3\right)}{100}$$
(4)OS=MUFA+45×C18:2+100×C18:3
(5)$$\frac{h}{H}=\frac{\left(C18:1+C18:2+C18:3\right)}{C16:0}$$
(6)PI=(C16:1t+C18:1)×0.025+C18:2+2×C18:3

### 2.3. Pigment Analysis

Pigments were extracted from 100 to 200 mg ground grapevine leaves using 100% cold acetone and maintained in a cold ultra-sound bath for 2 min. Samples were maintained in the dark overnight at −20 °C to continue extraction [40,41,42] and centrifuged at 4000 rpm for 15 min at 4 °C. The supernatants were collected and analysed using a dual-beam spectrophotometer (Shimadzu UV-1603). Absorbance spectrums were performed from 350 to 700 nm (0.5 nm steps) and introduced in the Gauss-Peak Spectra (GPS) fitting library, using SigmaPlot Software.

To detect chlorophyll (Chl) *a* and *b* and allomer (alloChl) *a*, pheophytin (Pheo) *a* and *b* and allomers (alloPheo) *a* and *b*, auroxanthin (Auro), antheraxanthin (Anth), β-carotene (β-Car), *trans*-lutein (Lut), neoxanthin (Neo), violaxanthin (Vx) and zeaxanthin (Zx) and isomers *trans*-Zeaxanthin (*trans*ZX), *Cis*-9-Zeaxanthin (*cis*9Zx), *Cis*-13-Zeaxanthin (*cis*13Zx), the algorithm developed by Küpper et al. (2007) [43] was applied.

### 2.4. Elemental Profiling

An oven-dried (60 °C until constant weight) sub-sample (≈100 mg dry weight) for leaf samples was mineralised using an acid mixture of HClO_4_:HNO_3_ (7:1 *v*/*v*) at 110 °C for 3 h in a closed Teflon vessel [44]. After cooling, the mineralisation products were filtered by Whatman 42 filter and added with an internal standard (Gallium, final concentration 1 mg/L) and stored at 4 °C until analysis. Elemental quantification was performed through Total X-ray Fluorescence spectroscopy (TXRF) element analysis, which is a reliable, environmentally friendly and efficient method [45]. The cleaning and preparation of the TXRF quartz glass sample carriers were performed according to Towett et al. (2013) [45]. For sample analysis, 5 μL of the digested sample was added to the centre of the quartz carrier and the carriers with the samples were aligned in a sample holder. Additionally, three internal calibration standards were also simultaneously analysed, containing a carrier with arsenic for gain correction (mono-element standard, Bruker Nano GmbH), a carrier with a nickel standard for sensitivity and detection limit (mono-element standard, Bruker Nano GmbH) and a carrier with a multi-element kraft for quantification accuracy (Kraft 10, Bruker Nano GmbH). Element measurements were made in a portable benchtop TXRF instrument (S2 PICOFOXTM spectrometer, Bruker Nano GmbH) for 1000 s per sample (Human et al., 2020). The accuracy and precision of the analytical methodology for elemental determinations were assessed by replicate analysis of certified reference material ERM-CD281 (Ryegrass reference material from Institute for Reference Materials and Measurements, reference ERM-CD281). All measured values were within the certified ones. The TXRF spectra and data evaluation interpretation were performed using the Spectra 7.8.2.0 software (Bruker Nano GmbH, Berlin, Germany).

### 2.5. Statistical Analysis

A multivariate approach was applied to the pigments and element analysis to evaluate the differences of these parameters among the genotypes. Primer 6 software was used to perform canonical analysis of principal coordinates (CAP) using Euclidean distances [46]. CAP allows to visualise differences in multivariate space as well as to determine how accurately samples could be allocated to different cultivars. CAP is insensitive to heterogeneous data and frequently used to compare different sample groups using the intrinsic characteristics of each group [41,47,48]. Due to the lack of normality and homogeneity of variances, statistical significance between pigment and element concentrations were performed by Kruskal–Wallis non-parametric test followed by a Dunn’s pairwise comparisons test, using IBM^®^ SPSS^®^ Statistics software (version 23.0; SPSS Inc., Chicago, IL, USA). Results with a *p*-value < 0.05 were considered statistically significant.

## 3. Results

### 3.1. Pigment Composition

Natural pigments are integral components of the plant with bioactive value as they have health-promoting properties [49]. Thus, seventeen pigments were analysed, namely the chlorophyll and carotenoid levels and variability. Chlorophylls a and b (from 230.69 ± 46.00 to 374.26 ± 34.95 µg/g and 198.53 ± 31.47 to 281.04 ± 17.48 µg/g, respectively) and zeaxanthin (from 11.42 ± 3.73 to 88.49 ± 18.34 µg/g) presented the highest concentration values (Table 2). A CAP was used to evaluate the ability to distinguish cultivars based on their pigment profiles. A 71.43% classification efficiency (Figure 1A) allowed to discriminate ‘Cabernet sauvignon’ due to its high levels of chlorophylls (Chl *a*, Chl *b* and alloChl *b*). Moreover, ‘Trincadeira’ and ‘C19’ stood out for their higher content in pheophytin (Pheo *a* and *b* and alloPheo *a* and *b*). The highest levels of β-carotene were observed in ‘Tinta barroca’, ‘Castelão’, ‘Bastardo’ and ‘C19’. Moreover, in ‘C19’, the content of *trans*-Lutein and zeaxanthin and derivatives was higher when compared with other genotypes (Table 2). The cultivar ‘Pinot noir’ appeared to be separated from all genotypes.

### 3.2. Elemental Profile

Chemical elements participate in a diverse range of biological functions, having a vital role in human physiology. Sodium (Na), calcium (Ca) and potassium (K) presented the highest concentrations in all genotypes. On the other hand, thallium (Ti) (0 to 2.39 ± 3.94 μg/g), vanadium (V) (0.03 ± 0.06 to 0.31 ± 0.04 μg/g), chromium (Cr) (0 to 0.04 ± 0.09 μg/g), cobalt (Co) (0.07 ± 0.06 to 0.20 ± 0.23 μg/g), nickel (Ni) (0.04 ± 0.05 to 0.13 ± 0.04 μg/g), bromine (Br) (0.14 ± 0.03 to 1.39 ± 0.40 μg/g), iodine (I) (0 to 1.20 ± 2.21 μg/g) and selenium (Se) (0 to 0.02 ± 0.02 μg/g) showed the lowest concentrations, not being detected in some cultivars (Table 3).

Iron (Fe) was also present in higher concentrations, compared to other genotypes, in ‘Castelão’ (29.26 ± 27.19 μg/g). The highest concentrations of zinc (Zn) were detected in *V. vinifera* cv. Pinot noir and ‘Cabernet sauvignon’ with, respectively, 30.70 ± 3.70 and 55.44 ± 5.41 ug/g.

The CAP analysis allowed an 88.57% classification efficiency of the grapevine cultivars based on their elemental profiling. ‘Cabernet sauvignon’ presented a high content in Zn. Calcium (Ca) and manganese (Mn) contribute to the grouping of ‘Tinta barroca’ and ‘Bastardo’ (Figure 1B). In addition, higher content in copper (Cu) contributes to ‘Pinot noir’ separation from the remaining cultivars. Through these results it could also be highlighted that three main groups can be separated considering the first canonical axis (CAP1): ‘Pinot noir’ and ‘Cabernet sauvignon’; ‘Trincadeira’, ‘Castelão’ and ‘C19’; and ‘Tinta barroca’ and ‘Bastardo’. Considering the second canonical axis (CAP2), ‘Tinta barroca’, ‘Bastardo’ and ‘Cabernet sauvignon’ can also be separated from the other four genotypes.

### 3.3. Fatty Acid Profile

Fatty acids have multiple roles in complex metabolic pathways and have a wide range of health-promoting properties [50]. Six FAs were detected: palmitic acid (C16:0), *trans*-hexadecaenoic acid (C16:1*t*), stearic acid (C18:0), oleic acid (C18:1), linoleic acid (C18:2) and α-linolenic acid (C18:3) (Table 4). A grapevine leaf FA profile was dominated by polyunsaturated FAs (PUFAs) (68.75%), represented by C18:2 and C18:3, followed by the saturated FAs (SFAs) (22.52%) C16:0 and C18:0. The monounsaturated FA (MUFAs) group was represented by C16:1*t* and C18:1 (8.73%).

Among the FAs detected, the omega-3 C18:3 was the most abundant, with almost half the total percentage of all FAs present in grapevine leaves (45.83%). The highest C18:3 concentration was detected in ‘Tinta barroca’. The omega-6 C18:2 was the second most abundant FA in grapevine leaves and its highest values were detected in ‘Pinot noir’ and ‘Cabernet sauvignon’.

Considering the role of PUFAs in cardiac diseases, two indexes, the atherogenicity (AI) and thrombogenicity (TI) indexes, are commonly used to infer and predict the potential health benefits associated with FA ingestion. The smaller the AI and TI values, the greater the protective potential for the cardiovascular system [35]. The oxididability (Cox) and oxidative susceptibility (OS) indicate the oxidative stability of FAs. While Cox values should be low, indicating that FAs are less prone to oxidation, OS should be as high as possible [51,52]. The ratio between hypocholesterolemic (h) and hypercholesterolemic (H) FA (h/H index) indicated the effects of specific FA on cholesterol metabolism. This index is of fundamental importance because the h reduces the low-density lipoproteins cholesterol, also known as bad cholesterol, whereas the H increases it [53].

The peroxidisability index (PI) is used to assess the stability of PUFAs and their capacity to be protected from possible oxidation processes. High h/H and PI index values are considered more beneficial for human health [53].

Our results highlighted that grapevine leaves possess a health-promoting lipid content. Both AI and TI values were low and consistent among all grapevine genotypes analysed: 0.29 ± 0.03 and 0.23 ± 0.02, respectively. The h/H values of grapevine leaves in the present study ranged from 3.47 to 4.15. Regarding lipid oxidation, OS values in grapevine leaves were high, above 5000, in all genotypes and Cox values were low and consistent in all *V. vinifera* cultivars. The PI value was around 100 in all genotypes.

## 4. Discussion

In recent years, a new trend of healthy nutrition has gained prominence. Moreover, general awareness on how products are cultivated and a higher demand for more organic farmed products has grown. In fact, the European Union and United Nations demand more sustainable agricultural approaches and improvements to consumer health (Goals of the 2030 Agenda for Sustainable Development) [8].

The grapevine’s market value is widely associated with grape and wine production. This crop is one of the most economically important worldwide, with an estimated 7.4 mha of area under vines which are destined for all purposes [1]. In 2018, 77.8 million tons of grapes were produced for wine production and commercialisation of fresh and dried grapes [54]. This also generates a considerable amount of waste. To facilitate vineyard management practices and the production of grapes with the desired quality, pruning is an essential procedure. During the growth season, this crop is regularly pruned, either manually or mechanistically, producing vegetative waste products that are often left in the vineyard or used to feed livestock [55]. As a result, the estimated amounts of material generated during pruning varied from 0.56 to 2.01 kg/vine depending on the trellis system, cultivar and year [2]. The major by-product generated from summer pruning is grapevine leaves. Very little attention has been paid to this by-product and it is normally disregarded. Although grape applications and health benefits have already been demonstrated [56,57], the investigation of the chemical characteristics of grapevine leaves is a major goal to evaluate this species as a source of natural beneficial compounds. Although they are considered a waste product, grapevine leaves are rich in essential nutrients, being an excellent source of bioactive compounds that exert several health beneficial properties. In some Mediterranean countries, such as Turkey and Greece, these leaves are used for human consumption, both fresh and brined [4,18,28,58] or used as a food supplement together with synthetic vitamins (E and C) and minerals (selenium) [21,59].

Hence, to improve the nutritional knowledge on this by-product, the elemental, FA and pigment composition were evaluated on seven *Vitis* genotypes (‘Trincadeira’, ‘Cabernet sauvignon’, ‘Pinot noir’, ‘Tinta barroca’, ‘Castelão’, ‘Bastardo’ and ‘C19’). All genotypes analysed in this study were from CAN. All plants were maintained with the same conditions: homogeneous modern alluvial soils (lowlands) as well-drained soil; rootstock of a unique variety (Selection Oppenheim 4–SO4); each accession comes from one unique plant; same trailing system (bilateral cordon, Royat), canopy maintenance and agricultural management. The climate of this region is temperate with a dry and mild summer in almost all regions of the northern mountain system Montejunto-Estrela and the regions of the west coast of Alentejo and Algarve.

Chemical elements play a wide range of important functions and are essential in human physiology [60,61,62]. Essential elements can be divided into two main groups: sodium (Na), potassium (K), calcium (Ca), magnesium (Mg) and phosphorus (P), which are needed in larger amounts by the human body, and copper (Cu), iron (Fe), manganese (Mn) and zinc (Zn), which are required in lower amounts [61,63]. In fact, elemental composition importance has already raised some new approaches in the viticulture sector. Some new breeding strategies focused on some elements’ accumulation are already being conducted. Recently, a leaf elemental profile was used to select three ‘Cabernet Franc’ clones with the optimal content of several elementals which are linked to several neurodegenerative disorders [64].

The World Health Organization (WHO) recommends that the daily intake of Na should not exceed 2000 mg. As for K, there is no upper limit established by the WHO but a K intake above 3500 mg/day is recommended [65]. For Ca, the daily intake should be above 1000 mg [66]. These three elements are the most abundant in all the tested grapevine cultivar leaves which is in agreement with other studies [21]. With the advice to have at least five servings of vegetables and/or fruits a day, considering 200 g of grapevine leaves, an intake of around 145 mg of Na, 700 mg of K and 430 mg of Ca is achieved. Hence, a serving of grapevine leaves corresponds to 7% of Na, 20% of K and 43% of Ca of the recommended daily consumption of these elements by the WHO. Moreover, compared to other vegetables (Table 5), the concentration of K in grapevine leaves is similar to what is found in asparagus, broccoli and cabbage. Calcium values are higher than in spinach and watercress. Moreover, grapevine leaves possess the highest levels of Na, except when compared with spinach.

Iron and Zn are also present in high concentrations in grapevine leaves, particularly in ‘Castelão’ and ‘Pinot noir’ and ‘Cabernet sauvignon’, respectively. Iron is highly present in human proteins, with the human genome coding for over 500 Fe containing proteins [67]. Low Fe intake may lead to anaemia, one of the most common health diseases in the world [68], thus a higher Fe intake has been associated with a positive role in human health [62]. Adult men need about 8 mg/day of iron to maintain balance [69]. However, the average requirement for women is higher: 14.8 mg/day. Moreover, during pregnancy, iron needs are higher, up to 27 mg/day. Requirements of iron vary according to age, being higher in childhood, particularly during periods of rapid growth in early childhood (6 to 24 months) and adolescence [69,70,71]. Taking this information into account, an intake of 200 g of the grapevine leaves in the study corresponds to a daily average of 3.75 mg of Fe. A similar value of Fe intake is achieved with a serving of watercress (2.6 mg). Due to the high need for Fe, grapevine leaves can also be a viable source of this element for human consumption or for Fe extraction to produce supplements.

Zn is involved in nearly all aspects of molecular and cell biology such as cellular growth and cellular differentiation and metabolism. One of its main roles is the stabilisation of proteins that are essential to the mammalian circadian clock [62]. Although severe zinc deficiency is rare in humans, mild to moderate deficiency may be common worldwide. Deficiency limits childhood growth and decreases resistance to infections. The recommended average daily intake of zinc is set at 15 mg/day for an adult man, 12 mg/day for adult women and 10 mg/day for preadolescent children [72]. An intake of 200 g of grapevine leaves gives around 4 mg of Zn, more than 1/3 of the daily recommended intake for men and 1/3 of the intake for adult women. Compared to other vegetables, grapevine leaves possess higher values except when compared with spinach (Table 5).

All the vegetables presented in Table 5 are well established in the human diet and some are considered to be among the top health-promoting vegetables. The element values obtained in our study are similar to these vegetables, demonstrating the potential of grapevine leaf consumption associated with several positive effects on human health.

Fatty acids are important nutrients, as they are sources of energy, precursors of biomolecules and important components of biological structures such as cell membranes [63,73]. An adult intake between 12–17 g of C18:2 and between 1.1–1.6 g of C18:3 [50,74] is recommended daily. Moreover, it is well known that the consumption of green-leafed vegetables reduces lipid peroxidation by 5%, reducing the risk of cardiovascular disease as a result of the improvement in endothelial function and blood pressure due to the antioxidants and nitrates the leaves contain [75].

In grapevine leaves, C18:3 was the predominant FA, accounting for 50% of the total FA content, followed by C18:2 and C16:0. The PUFAs/SFAs ratio (3) was highly above the minimum recommended (0.45) for suitable dietary sources of FA [39]. Additionally, all the leaves from the different genotypes contain a low percentage of MUFAs. In fact, the FA percentages in grapevine leaves are within the same values of other Mediterranean vegetables highly used for human consumption such as watercress (Table 6) [63,76,77].

Considering the potential contribution of PUFAs in cardiac diseases, two lipid-based indexes are commonly used to infer and predict the potential health benefits [35]. AI relates to the main classes of saturated (pro-atherogenic—favouring the adhesion of lipids to cells of the immunological and circulatory system) and unsaturated FAs (anti-atherogenic—inhibiting the aggregation of plaque and diminishing the levels of esterified FA, cholesterol and phospholipids, thereby preventing the appearance of micro coronary and macro coronary diseases). A low AI is recommended [35]. The TI index is the ratio between a prothrombogenic (saturated) and antithrombogenic FA and relates to the tendency to form clots in the blood vessels. Both indexes indicate the potential for stimulating platelet aggregation [39]. For human consumption, values below 1.0 (AI) and 0.5 (TI) are desirable [38]. AI and TI values in grapevine leaves are within the recommended values and are similar to other green leafy vegetables such as broccoli (AI: 0.41; TI: 0.18), carrot (AI: 0.25; TI: 0.38) and asparagus (AI: 0.36; TI: 0.37).

Concerning the other indexes, nutritionally higher h/H values are considered more beneficial for human health. The majority of the values found in grapevine leaves are higher than those found in macroalgae species, namely 1.26, 1.90, 2.09 and 4.22 for *U. rigida*, *U. compressa*, *P. capillacea* and *G. microdon*, respectively [53]. The OS, PI and Cox indexes are related to the susceptibility to oxidation of the FA. Grapevine leaf values for OS and Cox indexes were in accordance with the expected. In E. Covaci et al. [51], the authors refer that the oxidisability value (Cox) should be as low as possible, while oxidative susceptibility (OS) as high as possible. In fact, in their work they compare Cox and OS values from different oils and, for example, Lycium chinense M. oil has Cox and OS values of 6.93 and 2983, respectively; Croatian olive oil has values of 1.85 and 567; and Brazilian soybean oil has Cox and OS values ranging from 6.41–6.94 and 2744–2987, respectively. Our results are in accordance with these studies, leading us to speculate that FAs in grapevine leaves have a high oxidative protective potential. The PI value in grapevine leaves is similar to the ones found in different everyday dietary oils. For instance, soybean, perilla and fish oils have PI values of 70.78, 134.82 and 262.08, respectively. Although it is described in the literature that the higher the PI values, the greater the susceptibility to lipid peroxidation, Min Jeong Kang and co-workers proved that a high polyunsaturated/saturated fatty acid ratio diet has a beneficial effect on cardiovascular disease risk even without antioxidant when the PI value is the same, above 80 [78]. This is because a high P/S ratio diet makes it difficult to increase lipid peroxidation because of the high concentrations of PUFA [79]. There are very few studies about what the optimal PI value might be hence further studies are encouraged to evaluate this index in the everyday diet.

**Table 5 foods-10-02251-t005:** Element composition in different plants in comparison with grapevine leaves.

Elements (μg/g)	Mean Values of Grapevine Leaves	Asparagus [80,81]	Broccoli [82]	Carrot [83]	Cabbage [84]	Chinese Cabbage [84]	Red Cabbage [84]	Savoy Cabbage [84]	Spinach [75]	Watercress [63]
**Na**	724.00 ± 152.33	32	360	NA	500	650	270	280	1200	130
**Mg**	21.34 ± 32.59	170	210	90	1300	190	160	280	580	270
**K**	3515.05 ± 626.60	3200	3030	2400	34,500	2520	2430	2300	6330	2740
**Ca**	2139.97 ± 275.87	360	460	340	5600	1050	450	350	1260	1510
**Mn**	10.74 ± 0.00	2.100	1.970	NA	1.600	1.590	2.430	1.800	87.500	2
**Fe**	18.77 ± 6.03	6.300	6.900	4	53170	8	8	4	40 to 350	13
**Cu**	3.28 ± 0.27	2.100	0.590	0.200	0.190	0.210	0.170	0.620	1.280	2
**Zn**	19.89 ± 2.34	7.600	4.200	2	1.800	1.900	2.200	2.7	5 to 42.500	5
**Se**	0.01 ± 0.02	0.060	0.016	NA	3	0.005	0.006	0.009	NA	NA

**Table 6 foods-10-02251-t006:** Fatty acid percentage in different plants in comparison with grapevine leaves.

FA	Mean Values of Grapevine Leaves	Asparagus [80,81]	Broccoli [82]	Carrot [83]	Cabbage [84]	Chinese Cabbage [84]	Red Cabbage [84]	Savoy Cabbage [84]	Spinach [75]	Watercress [63]
**PUFAs (%)**	69.06 ± 3.63	37.0	25.4	69.9	25.0	69.6	70.8	64.5	64.5	81.2
**MUFAs (%)**	7.77 ± 1.83	22.2	16.4	8.2	25.0	10.9	10.1	14.4	14.4	2.1
**SFAs (%)**	23.17 ± 1.91	40.7	58.2	21.9	50.0	19.6	18.8	21.1	21.1	16.7

Natural plant chlorophyll derivatives are mainly used as food colorants but have recently gained prominence as bioactive or functional compounds as they present anti-inflammatory activity in vitro [49,85,86] and contribute to the prevention of cancer and cardiovascular dysfunctions [87]. The regular ingestion of fruits and vegetables contributes to healthy chlorophyll and carotenoid levels. Moreover, the ingestion of seaweeds, microalgae, functional drinks and food supplements allows the increase in these nutrient uptakes and has been linked to new and healthy trends in feeding habits. Particularly, carotenoids and, to a lesser extent, chlorophyll have been shown to decrease oxidation of other molecules and thus are considered as antioxidant compounds.

Carotenoids are well recognised for their antioxidant capacity. β-carotene and lutein are two of the most prevalent carotenoids in the human diet and both pigments serve as antioxidants scavenging free radicals and quenching singlet oxygens [88]. Lutein has also been linked to having a protective effect against the risk of eye diseases, for example, cataracts and macular degradation [89,90]. Lutein is the stereoisomer of zeaxanthin and both carotenoids are generally found together in plant sources [89]. The levels of specific pigments vary between plants and within the same plant species. In fact, through the years they have been extracted from plants and by-products to be introduced in the diet as supplements or ingredients.

In grapevine leaves, the highest levels of pigments were detected for chlorophylls, β-carotene, *trans*-lutein and zeaxanthin. Chlorophylls were the most abundant pigments in ‘Cabernet sauvignon’. Interestingly, ‘C19’ presents higher values of pheophytin, β-carotene, *trans*-lutein, zeaxanthin and derivatives when compared to the other genotypes. ‘C19’ has a higher content of β-carotene (29.42 ± 4.60 μg/g FW), when compared to red cabbage (6.7 μg/g FW), savoy cabbage (6 μg/g FW) and Chinese cabbage (25.46 μg/g FW) [84]. Nevertheless, ‘C19’ levels of β-carotene are considered low when compared to dark orange carrots (170 μg/g) and spinach (56 μg/g) [75,83]. In ‘C19’, the values of *trans*-lutein (34.63 ± 5.33 μg/g) are also higher than white (0.30 μg/g), savoy (0.77 μg/g) and red cabbages (3.29 μg/g) and yellow carrots (5.05 μg/g) [84,91]. Interestingly, zeaxanthin values in ‘C19’ (88.498 ± 18.34 μg/g) are higher than the values observed for spinach (1.35 μg/g).

This cultivar is a Portuguese breed mainly developed by Coutinho [31,32] to possess resistance to pathogens. Nowadays, the majority of grapevine breeding strategies focus on the creation of new cultivars which possess high levels of tolerance towards diseases and also desirable characteristics for wine production. However, in other crops, some pigments have been targeted for breeding strategies [92]. Our results demonstrate that a thorough investigation on grapevine cultivars can also reveal other applications besides wine.

Another point to have in consideration in the inclusion of grapevine leaves in the human diet or as a source for the extraction of bioactive compounds is the environment where grapevines are planted and the way they are processed. The composition and concentration of bioactive components may vary with the genotype, the environmental condition in which the crop grows, harvesting time and conditions, handling and storage, industrial food processing and meal preparation. The plant’s age and reproductive status also affect the content of phytochemicals in plants.

The genotype variation is also a prime factor known to directly influence the phytochemical composition of all living organisms as it is what determines their individual characteristics in terms of their nutrient content as well as colour, shape and size. In fact, as it was observed by our results, element, FA and pigment composition varied depending on the genotype. These bioactive compound differences in genotypes were also reported in other crops such as cabbage and carrots [83,84].

Since the grapevine is such a diverse crop and has many genotypes, some of which are more common in certain parts of the world/countries than others, it would be interesting to investigate how the element, FA and pigment composition vary upon different locations and through grapevine season. Moreover, to evaluate their content profiles in coastal versus inland locations.

## 5. Conclusions

The wine industry produces an enormous amount of waste every year. Due to current demands from different organisations for more sustainable agricultural practices, it is urgent to add value to this industry’s by-products. In this work, we have shown the high potential of grapevine leaves for nutrition and as a source of bioactive compounds. We have evaluated elemental, FA and pigment profiles on seven different genotypes with high economic value. We have shown that grapevine leaves possess a health-promoting lipid content. All the values of the FA-derived parameters analysed in this study are in accordance with the recommended values for human consumption. Ingestion of a small portion of grapevine leaves was also shown to provide the amount of several elements that enable to achieve the WHO’s daily recommended doses. Regarding the pigment content, ‘C19’ presented the highest values when compared to the other genotypes, showing the highest levels of pheophytin, β-carotene, *trans*-lutein and zeaxanthin and derivatives.

In view of all the above, more research is encouraged to be undertaken regarding the various genotypes and breeding practices of the grapevine, as well as the different climatic factors, in order to fully understand the phytochemical composition of grapevine leaves and all its potential to be used in human consumption.

## Figures and Tables

**Figure 1 foods-10-02251-f001:**
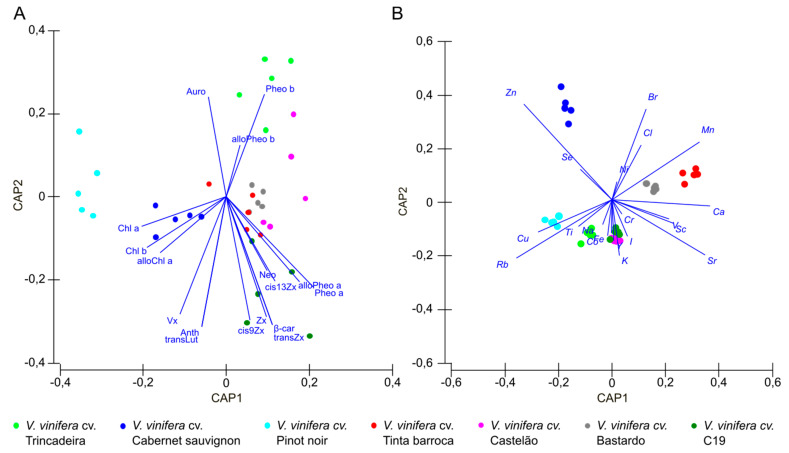
Canonical analysis of principal coordinates (CAP) plots between *Vitis* genotypes considering (**A**) pigments concentration and (**B**) the elements leaf concentration.

**Table 1 foods-10-02251-t001:** *Vitis vinifera* cultivars selected for analyses. Species and cultivar names, VIVC variety number, type of accession, origin and colour of berry skin.

	Subspecies (Subsp.) and/or Cultivar (cv.)	VIVC Variety Number	Type of Accession	Origin	Colour of Berry Skin
** *Vitis vinifera* ** **cv.**	Subsp. sativa cv. Trincadeira	15,685	Unknown Natural hybridisation	Portugal	Noir
	Subsp. sativa cv. Cabernet sauvignon	1929	Natural hybridisation Cabernet Franc X Sauvignon	France	Noir
	Subsp. sativa cv. Pinot Noir	9279	Savagnin Blanc X?	France	Noir
	Subsp. sativa cv. Tinta Barroca	12,462	Natural hybridisation Marufo X Touriga Nacional	Portugal	Noir
	Subsp. sativa cv. Castelão	2324	Natural hybridisation Cayetana Blanca X Alfrocheiro Preto	Portugal	Noir
	Subsp. sativa cv. Bastardo	12,668	Natural hybridisation (? X Savagnin = Traminer	Portugal	Noir
	C19 *	Not registered on VIVC	Hybrid crossing (Jaen T X?) X Jaen T)	Portugal	Noir

* C19 T—Hybrid crossing obtained by Pereira Coutinho as a result of crossing (Jaen T X Azal Blanco B) X Jaen T varieties. This breeding line was created at Instituto Superior de Agronomia, Lisbon University, Portugal [31,32].

**Table 2 foods-10-02251-t002:** Average concentration and standard deviation of each pigment in grapevine leaves. N = 5, for a given row, different letters indicate significant differences at *p* < 0.05; same letters indicate no significant differences (*p* > 0.05) (Kruskal–Wallis test).

	*Vitis vinifera* cv.						
Pigment (µg/g FW)	Trincadeira	Cabernet Sauvignon	Pinot Noir	Tinta Barroca	Castelão	Bastardo	C19
Chlorophyll *a*	230.69 ± 46.00 ^a^	373.49 ± 35.05 ^b^	374.26 ± 34.95 ^b^	254.09 ± 65.26 ^ab^	283.53 ± 67.89 ^ab^	318.29 ± 35.10 ^ab^	269.32 ± 74.30 ^ab^
Chlorophyll *b*	203.95 ± 30.35 ^a^	281.04 ± 17.48 ^a^	276.95 ± 16.52 ^a^	198.53 ± 31.47 ^a^	214.81 ± 33.91 ^a^	211.28 ± 20.01 ^a^	259.17 ± 54.30 ^a^
Pheophytin *a*	27.62 ± 8.60 ^c^	33.58 ± 13.87 ^abc^	13.83 ± 12.58 ^ac^	57.96 ± 9.53 ^bc^	51.20 ± 13.19 ^abc^	48.69 ± 8.10 ^abc^	75.39 ± 17.21 ^b^
Pheophytin *b*	20.34 ± 14.17 ^a^	3.38 × 10^−23^ ± 6.05 × 10^−23 bcde^	5.22 × 10^−24^ ± 7.67 × 10^−24 bcde^	0.16 ±0.35 ^abcde^	3.44 ± 6.07 ^abcde^	6.45 × 10^−24^ ± 1.06 × 10^−23 bcde^	1.12 × 10^−23^ ± 1.94 × 10^−23 bcde^
alloChlorophyll *a*	65.19 ± 23.48 ^ab^	120.56 ± 35.32 ^b^	116.10 ± 28.55 ^b^	75.40 ± 20.53 ^ab^	59.89 ± 40.98 ^ab^	37.26 ± 11.78 ^a^	117.88 ± 29.52 ^b^
alloPheophytin *a*	35.74 ± 30.92 ^a^	63.33 ± 40.74 ^ab^	28.33 ± 42.46 ^ab^	102.95 ± 13.09 ^ab^	95.66 ± 11.46 ^ab^	111.80 ±15.26 ^b^	121.79 ± 29.56 ^b^
alloPheophytin *b*	3.92 ± 8.77 ^a^	3.67 × 10^−23^ ± 7.66 × 10^−23 a^	3.25 × 10^−24^ ± 3.05 × 10^−24 a^	3.05 × 10^−24^ ± 4.43 × 10^−24 a^	1.41 × 10^−24^ ± 1.47 × 10^−24 a^	1.56 × 10^−23^ ± 1.28 × 10^−23 a^	5.43 × 10^−24^ ± 7.65 × 10^−24 a^
Auroxanthin	61.48 ± 8.17 ^b^	27.51 ± 18.36 ^ab^	55.24 ± 12.02 ^ab^	34.16 ± 15.44 ^ab^	48.22 ± 27.32 ^ab^	45.44 ± 20.01 ^ab^	9.07 ± 10.40 ^a^
Antheraxanthin	10.61 ± 3.10 ^a^	27.57 ± 4.37 ^ab^	27.07 ± 6.68 ^ab^	26.56 ± 2.57 ^ab^	21.44 ± 4.52 ^ab^	25.18 ± 2.21 ^ab^	35.60 ± 5.48 ^b^
β-carotene	7.10 ± 2.70 ^a^	16.99 ± 2.29 ^ab^	12.46 ± 4.48 ^a^	20.38 ± 2.51 ^ab^	18.19 ± 3.22 ^ab^	20.32± 1.29 ^ab^	29.42 ± 4.56 ^b^
Trans-Lutein	10.32 ± 3.02 ^a^	26.81 ± 4.25 ^ab^	26.33 ± 6.50 ^ab^	25.83 ± 2.50 ^ab^	20.86 ± 4.40 ^ab^	24.49 ± 2.15 ^ab^	34.63 ± 5.33 ^b^
Neoxanthin	5.50 ± 4.38 ^a^	6.62 ± 7.59 ^a^	9.16 ± 7.91 ^a^	22.10 ± 3.99 ^a^	13.87 ± 8.36 ^a^	14.70 ± 3.16 ^a^	23.24 ± 10.96 ^a^
Violaxanthin	11.56 ± 4.42 ^a^	32.80 ± 7.94 ^abc^	35.07 ± 8.79 ^bc^	30.39 ± 2.17 ^abc^	22.50 ± 5.74 ^ab^	28.47 ± 3.15 ^abc^	40.58 ± 7.84 ^c^
Trans-Zeaxanthin	7.52 ± 2.86 ^a^	18.00 ± 2.42 ^ab^	13.20 ± 4.75 ^a^	21.59 ± 2.66 ^ab^	19.27 ± 3.41 ^ab^	21.54 ± 1.36 ^ab^	31.17 ± 4.87 ^b^
Cis-9-Zeaxanthin	0.58 ± 1.30 ^a^	16.65 ± 5.20 ^abc^	10.09 ± 7.69 ^ab^	22.41 ± 7.06 ^bc^	12.12 ± 12.32 ^abc^	15.82 ± 7.02 ^abc^	35.37 ± 5.68 ^c^
Cis-13-Zeaxanthin	3.31 ± 3.04 ^ab^	5.41 ± 5.44 ^ab^	1.78 ± 3.21 ^a^	14.17 ± 8.98 ^ab^	7.61 ± 10.46 ^ab^	5.01 ± 5.53 ^ab^	21.95 ± 10.08 ^b^
Zeaxanthin	11.42 ± 3.73 ^a^	40.05 ± 8.89 ^abc^	25.07 ± 13.44 ^ab^	58.17 ± 17.87 ^bc^	39.00 ± 25.37 ^abc^	42.37 ± 12.90 ^abc^	88.49 ± 18.34 ^c^

**Table 3 foods-10-02251-t003:** Average concentration and standard deviation of each element analysed through total reflection X-ray fluorescence (TXRF) in grapevine leaves. N = 5, for a given row, different letters indicate significant differences at *p* < 0.05; same letters indicate no significant differences (*p* > 0.05) (Kruskal–Wallis test).

	*Vitis vinifera* cv.						
Element (μg/g FW)	Trincadeira	Cabernet Sauvignon	Pinot Noir	Tinta Barroca	Castelão	Bastardo	C19
Na	491.72 ± 139.27 ^a^	661.66 ± 74.64 ^ab^	898.26 ± 288.80 ^ab^	746.88 ± 74.99 ^ab^	1033.40 ± 194.20 ^b^	571.39 ± 198.90 ^ab^	671.68 ± 95.52 ^ab^
Mg	21.59 ± 33.32 ^a^	14.64 ± 32.72 ^a^	28.61 ± 39.30 ^a^	0.00 ^a^	55.58 ± 57.97 ^a^	18.51 ± 41.39 ^a^	10.48 ± 23.44 ^a^
Cl	21.27 ± 5.69 ^abc^	29.01 ± 8.31 ^bc^	9.67 ± 6.19 ^a^	23.84 ± 6.31 ^abc^	9.25 ± 3.28 ^a^	25.08 ± 8.71 ^abc^	30.02 ± 7.51 ^c^
K	4046.22 ± 821.37 ^a^	2817.77 ± 336.38 ^a^	3446.68 ± 1054.71 ^a^	3224.14 ± 252.62 ^a^	3818.64 ± 1174.17 ^a^	3784.25 ± 552.03 ^a^	3467.66 ± 194.97 ^a^
Ca	2080.56 ± 392.42 ^abc^	1691.01 ± 106.95 ^a^	1580.98 ± 431.74 ^a^	2771.00 ± 327.37 ^c^	1752.66 ± 379.56 ^abc^	2513.26 ± 143.36 ^abc^	2590.35 ± 149.68 ^bc^
Sc	5.84 ± 1.41 ^abc^	4.61 ± 0.63 ^ab^	3.59 ± 0.82 ^a^	6.65 ± 0.72 ^bc^	4.86 ± 0.74 ^abc^	6.02 ± 1.10 ^abc^	8.65 ± 2.17 ^c^
Ti	0.26 ± 0.25 ^a^	0.12 ± 0.17 ^a^	2.39 ± 3.94 ^a^	0.27 ± 0.28 ^a^	1.10 ± 0.92 ^a^	0.00 ^a^	0.30 ± 0.31 ^a^
V	0.17 ± 0.11 ^ab^	0.12 ± 0.12 ^ab^	0.03 ± 0.06 ^a^	0.26 ± 0.16 ^ab^	0.19 ± 0.13 ^ab^	0.150 ± 0.102 ^ab^	0.31 ± 0.04 ^b^
Cr	0.00 ^a^	0.00 ^a^	0.02 ± 0.05 ^a^	0.03 ± 0.06 ^a^	0.04 ± 0.09 ^a^	0.00 ^a^	0.00 ^a^
Mn	9.30 ± 1.81 ^abc^	11.77 ± 0.69 ^abcd^	6.50 ± 0.80 ^a^	13.01 ± 1.48 ^cd^	9.08 ± 1.29 ^abc^	13.24 ± 0.50 ^d^	12.28 ± 0.42 ^bcd^
Fe	19.79 ± 6.83 ^a^	15.99 ± 1.62 ^a^	14.85 ± 3.16 ^a^	15.32 ± 1.06 ^a^	29.26 ± 27.19 ^a^	15.40 ± 1.01 ^a^	20.77 ± 1.33 ^a^
Co	0.16 ± 0.10 ^a^	0.07 ± 0.06 ^a^	0.10 ± 0.10 ^a^	0.06 ± 0.08 ^a^	0.20 ± 0.23 ^a^	0.15 ± 0.05 ^a^	0.10 ± 0.07 ^a^
Ni	0.13 ± 0.04 ^a^	0.11 ± 0.03 ^a^	0.10 ± 0.09 ^a^	0.11 ± 0.07 ^a^	0.04 ± 0.05 ^a^	0.12 ± 0.05 ^a^	0.12 ± 0.04 ^a^
Cu	3.26 ± 0.66 ^ab^	3.46 ± 0.11 ^ab^	3.69 ± 0.18 ^ab^	2.47 ± 0.11 ^a^	3.06 ± 0.38 ^ab^	2.49 ± 0.18 ^a^	4.51 ± 0.30 ^b^
Zn	11.16 ± 2.04 ^ab^	55.44 ± 5.41 ^b^	30.70 ± 3.70 ^ab^	9.41 ± 0.92 ^a^	9.53 ± 2.43 ^a^	9.31 ± 1.23 ^a^	13.66 ± 0.65 ^ab^
Se	0.00 ± 0.00 ^a^	0.02 ± 0.02 ^a^	0.02 ± 0.02 ^a^	0.00 ^a^	0.01 ± 0.03 ^a^	0.02 ± 0.04 ^a^	0.00 ^a^
Br	0.45 ± 0.10 ^abcd^	1.39 ± 0.40 ^d^	0.14 ± 0.03 ^a^	1.18 ± 0.071 ^cd^	0.23 ± 0.09 ^abc^	0.61 ± 0.38 ^abcd^	0.97 ± 0.43 ^bcd^
Rb	6.57 ± 1.46 ^cd^	5.18 ± 0.29 ^abcd^	7.16 ± 1.00 ^d^	2.90 ± 0.15 ^a^	5.83 ± 1.31 ^abcd^	4.57 ± 0.98 ^abcd^	6.04 ± 0.52 ^bcd^
Sr	4.79 ± 1.05 ^abc^	3.46 ± 0.35 ^a^	4.33 ± 0.56 ^ab^	6.39 ± 0.36 ^c^	5.71 ± 1.02 ^abc^	5.66 ± 0.96 ^abc^	6.31 ± 0.35 ^bc^
Y	2.84 ± 0.64 ^ab^	2.88 ± 0.51 ^ab^	4.22 ± 0.91 ^ab^	3.89 ± 0.60 ^ab^	4.29 ± 0.51 ^b^	2.72 ± 0.45 ^a^	3.37 ± 0.31 ^ab^
I	0.34 ± 0.54 ^a^	0.00 ^a^	0.27 ± 0.59 ^a^	0.56 ± 1.24 ^a^	1.20 ± 2.21 ^a^	0.20 ± 0.44 ^a^	1.11 ± 0.63 ^a^

**Table 4 foods-10-02251-t004:** Grapevine leaves FA composition: polyunsaturated FA (PUFAs); saturated FA (SFAs); monounsaturated FA (MUFAs); palmitic acid (C16:0); *trans*-hexadecaenoic acid (C16:1*t*); stearic acid (C18:0); oleic acid (C18:1); linoleic acid (C18:2); α-linolenic acid (C18:3); and indexes of lipid nutritional quality: atherogenic index (AI), thrombogenic index (TI), oxidisability (Cox), oxidative susceptibility (OS), hypocholesterolemic/hypercholesterolemic (h/H) and peroxidisability (PI). N = 5.

	*Vitis vinifera* cv.						
Fatty Acids and Derived Parameters	Trincadeira	Cabernet Sauvignon	Pinot Noir	Tinta Barroca	Castelão	Bastardo	C19
C16:0 (%)	21.22 ± 1.73	20.25 ± 0.74	21.36 ± 1.69	20.25 ± 0.59	19.21 ± 1.39	19.92 ± 2.09	17.92 ± 0.94
C16:1*t* (%)	2.64 ± 0.22	2.38 ± 0.12	2.78 ± 0.30	2.79 ± 0.20	2.89 ± 0.35	5.62 ± 1.51	3.96 ± 1.37
C18:0 (%)	1.95 ± 0.32	2.91 ± 1.94	2.28 ± 0.73	1.64 ± 0.28	2.17 ± 0.60	2.61 ± 0.93	3.97 ± 1.73
C18:1 (%)	5.14 ± 1.99	5.17 ± 0.46	5.33 ± 0.60	4.05 ± 1.14	5.40 ± 1.93	5.94 ± 1.98	6.99 ± 2.50
C18:2 (%)	23.71 ± 3.87	24.51 ± 1.06	24.71 ± 1.35	21.15 ± 3.12	20.91 ± 3.32	22.56 ± 0.73	22.88 ± 0.61
C18:3 (%)	45.35 ± 7.39	44.78 ± 1.90	43.53 ± 2.13	50.13 ± 4.24	49.42 ± 5.26	43.34 ± 2.90	44.27 ± 5.45
PUFAS (%)	69.06 ± 3.63	69.29 ± 2.72	68.24 ± 2.16	71.28 ± 1.14	70.33 ± 2.19	65.90 ± 2.59	67.15 ± 6.03
SFA (%)	23.17 ± 1.91	23.16 ± 2.43	23.65 ± 1.93	21.89 ± 0.42	21.38 ± 1.27	22.54 ± 1.52	21.89 ± 2.31
MUFAS (%)	7.77 ± 1.83	7.55 ± 0.35	8.11 ± 0.41	6.84 ± 0.97	8.29 ± 1.63	11.56 ± 3.04	10.95 ± 3.87
PUFA/SFA (%)	3.01 ± 0.41	3.03 ± 0.39	2.91 ± 0.36	3.26 ± 0.10	3.30 ± 0.28	2.93 ± 0.22	3.11 ± 0.56
AI	0.30 ± 0.03	0.30 ± 0.04	0.31 ± 0.03	0.28 ± 0.01	0.27 ± 0.02	0.29 ± 0.03	0.28 ± 0.04
TI	0.15 ± 0.03	0.15 ± 0.02	0.16 ± 0.02	0.13 ± 0.01	0.13 ± 0.02	0.15 ± 0.01	0.15 ± 0.03
Cox	12.29 ± 1.19	12.25 ± 0.49	12.00 ± 0.44	13.05 ± 0.58	12.88 ± 0.79	11.74 ± 0.57	11.99 ± 1.21
OS	5609.32 ± 566.21	5588.91 ± 224.36	5473.05 ± 202.84	5971.40 ± 282.62	5891.16 ± 380.86	5360.71 ± 271.28	5467.96 ± 567.58
h/H	3.52 ± 0.37	3.68 ± 0.24	3.47 ± 0.40	3.72 ± 0.12	3.96 ± 0.35	3.64 ± 0.40	4.15 ± 0.38
PI	114.60 ± 10.94	114.27 ± 4.57	111.97 ± 4.06	121.58 ± 5.34	119.96 ± 7.30	109.53 ± 5.38	111.70 ± 11.39

## Data Availability

Not applicable.

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
