# Peer review of "More than Just Wine: The Nutritional Benefits of Grapevine Leaves"

_foods, 2021, doi:10.3390/foods10102251_

Round 1

Reviewer 1 Report

The manuscript by Maia et al. reports an investigation of fatty acids composition, elemental profiling, pigments, and nutritional quality indexes of leaves from different grapevine cultivars. These leaves can be considered waste but they are sometimes included in the human diet. This inclusion is interesting both from an environmental and a nutritional point of you, as the authors clearly pointed out in the introduction. 

The methods are well described, the study was well designed, and had an appropriate methodology. Overall, the results are extremely interesting. I agree that further experiments are required, exploring various genotypes, breeding practices of grapevine, and climatic factors. Anyway, they lead the base for future applications.

Some minor comments:

  • Line145: put the number in the formula as a subscript. 
  • Line 205: ug/g should be µg/g.
  • Table 2: ug/g should be µg/g.
  • Line 439: delete this extra line.

Author Response

Review Report Form

Reviewer 1:

The manuscript by Maia et al. reports an investigation of fatty acids composition, elemental profiling, pigments, and nutritional quality indexes of leaves from different grapevine cultivars. These leaves can be considered waste but they are sometimes included in the human diet. This inclusion is interesting both from an environmental and a nutritional point of you, as the authors clearly pointed out in the introduction.

The methods are well described, the study was well designed, and had an appropriate methodology. Overall, the results are extremely interesting. I agree that further experiments are required, exploring various genotypes, breeding practices of grapevine, and climatic factors. Anyway, they lead the base for future applications.

The authors acknowledge the reviewers’ positive comments to the manuscript submitted. The minor changes requested by the reviewer were performed in the manuscript.

Some minor comments:

Line145: put the number in the formula as a subscript.

The numbers present in the formula were put as a subscript as requested by the reviewer.

Line 205: ug/g should be µg/g.

This was corrected in the manuscript.

Table 2: ug/g should be µg/g.

This was corrected in the manuscript.

Line 439: delete this extra line.

This line was deleted.

Reviewer 2 Report

  1. The standard deviation is too huge, thus affects the significant difference in the comparison among different samples. Therefore, judgement of the results is affected. The test should be repeated.

  1. Authors did not mark the significant difference in Table S1 (Chlorophyll b, Neoxanthin, alloPheophytin b) and Table 2 (Mg, K, Ti, Fe, Co, Ni, Se, I). Does this mean that the authors did not perform statistic?

  1. Refer to the formula in Line 124, 125, 127, and reference [27]: “The PI index is used to assess the stability of PUFA included in food products” and “Lower PI value in W31 indicates a lower level of fatty acids autooxidation”, PI, Cox and OS values are referred to the susceptibility of fatty acids to oxidation, which indicate the stability of fatty acids. However, the authors had given contradiction statement. In Line 253-257, the authors stated “Oxidative susceptibility values in grapevine leaves are very high (above 5000 in all genotypes), being this index is in agreement with the PI values which are above 100 in all V. vinifera cultivars, indicating that the FAs present in grapevine leaves are stable and do not oxidate easily. On the contrary, Cox values are low and consistent in all Vitis genotypes” and in Line 380-382, the authors mentioned that: “Grapevine leaves values for these indexes were according to the expected leading us to speculate that FA in grapevine leaves are no easily oxidated.”

  1. Refer to the article written by B-J, Lee; et al, “Effects of the P/S Ratio of Dietary Lipids and Antioxidant Vitamin Supplements on the Level of Serum Lipids and Liver Lipid Peroxidation in Rats Treated with DMBA” published in the Korean Journal of Nutrition 31(5): 906-913, 1998, “high intake of unsaturated fat was effective on reducing total cholesterol and triglyceride concentration in serum, but resulted in the increase of unsaturation of fatty acid profile in serum and lipid peroxidation in liver. Consequently increased the need to reduce the formation and accumulation of the free radicals and lipid peroxides. When polyunsaturated fat intake increased drastically, to protect from lipid peroxidation, the function of antioxidant vitamins became more important …” In Reference [27] as well, the author stated that “The excessive intake of PUFA has undesirable effects such as oxidative stress because of high susceptibility to lipid peroxidation. Oxidative stress, which is associated with the formation of lipid peroxides, has been suggested as contributing to pathological processes in aging and many diseases such as atherosclerosis.” At this context, why did the authors think that higher values of PI, Cox and OS are more beneficial for human health (Line 249)?

  1. Line 256:Authors stated that Cox values are low, however, if PI and OS values are

high, Cox values should be high as well. What are the standard values that the authors

referred to decide whether the values are high or low?

  1. Line 439: the sentence is not related to this study

  1. Line 24 and 26: V. vinifera cv. should be V. vinifera cv.

Author Response

Reviewer 2:

  1. The standard deviation is too huge, thus affects the significant difference in the comparison among different samples. Therefore, judgement of the results is affected. The test should be repeated.

The authors acknowledge the reviewer comments. This preliminary approach used field grown material, thus genotypic and environmental influence was expected. Biological replication was conducted and despite the fact some standard deviations are high, conclusions were drawn based on statistics or evident tendencies. Our main aim was to characterize the fatty acid, elements and pigment content of grapevine leaves and to show their nutritional potential and this was achieved for all the genotypes studied. Even though that not very significant differences were found within cultivars, our main aim was to exploit the potential of grapevine leaf use and we consider that the overall aim of this study was achieved. Thus, we are certain that the main conclusions are supported by the data obtained and find no need to repeat this experiment.

Moreover the authors refer throughout the manuscript text that more studies have to be conducted namely considering different seasons and several collections within the same season.

  1. Authors did not mark the significant difference in Table S1 (Chlorophyll b, Neoxanthin, alloPheophytin b) and Table 2 (Mg, K, Ti, Fe, Co, Ni, Se, I). Does this mean that the authors did not perform statistic?

The authors performed a statistical analysis in all pigments and elements. However, no significant differences were observed in Mg, K, Ti, Fe, Co, Ni, Se, I (Table 2) and Chlorophyll b, alloPheophytin b and Neoxanthin (Table S1) in the leaves of the grapevine genotypes tested. Thus, in the text only general tendencies are referred.

  1. Refer to the formula in Line 124, 125, 127, and reference [27]: “The PI index is used to assess the stability of PUFA included in food products” and “Lower PI value in W31 indicates a lower level of fatty acids autooxidation”, PI, Cox and OS values are referred to the susceptibility of fatty acids to oxidation, which indicate the stability of fatty acids. However, the authors had given contradiction statement. In Line 253-257, the authors stated “Oxidative susceptibility values in grapevine leaves are very high (above 5000 in all genotypes), being this index is in agreement with the PI values which are above 100 in all V. vinifera cultivars, indicating that the FAs present in grapevine leaves are stable and do not oxidate easily. On the contrary, Cox values are low and consistent in all Vitis genotypes” and in Line 380-382, the authors mentioned that: “Grapevine leaves values for these indexes were according to the expected leading us to speculate that FA in grapevine leaves are no easily oxidated.”

The authors acknowledge the reviewer comments. The indexes pointed out in the manuscript are used to assess the stability of FA in food products and their protection capacity from possible oxidation processes. Higher OS and PI values and lower Cox values are associated to a greater protective potential (DOI:10.3233/JBR-190450; https://doi.org/10.1016/j.psj.2019.10.026).

As the sentence referred may lead to misinterpretations by the readers, it was changed as follows:

Page 8, line 274-278: “Regarding lipid oxidation, FAs present in grapevine leaves are stable and do not oxidize easily as OS and PI values in grapevine leaves are high, above 5000 and 100 in all genotypes, respectively.  In addition, Cox values are low and consistent in all V. vinifera cultivars. In fact, higher values of OS and PI as well as lower values of Cox represent less lipid susceptibility to oxidation.”    

  1. Refer to the article written by B-J, Lee; et al, “Effects of the P/S Ratio of Dietary Lipids and Antioxidant Vitamin Supplements on the Level of Serum Lipids and Liver Lipid Peroxidation in Rats Treated with DMBA” published in the Korean Journal of Nutrition 31(5): 906-913, 1998, “high intake of unsaturated fat was effective on reducing total cholesterol and triglyceride concentration in serum, but resulted in the increase of unsaturation of fatty acid profile in serum and lipid peroxidation in liver. Consequently increased the need to reduce the formation and accumulation of the free radicals and lipid peroxides. When polyunsaturated fat intake increased drastically, to protect from lipid peroxidation, the function of antioxidant vitamins became more important …” In Reference [27] as well, the author stated that “The excessive intake of PUFA has undesirable effects such as oxidative stress because of high susceptibility to lipid peroxidation. Oxidative stress, which is associated with the formation of lipid peroxides, has been suggested as contributing to pathological processes in aging and many diseases such as atherosclerosis.” At this context, why did the authors think that higher values of PI, Cox and OS are more beneficial for human health (Line 249)?

As indicated in reference [27], lower PI values indicate a lower level of fatty acids auto oxidation and higher values of the PI index may indicate a higher prohealth value.

In E. Covaci et al. (DOI:10.3233/JBR-190450), the authors refer that the oxidisability value (Cox) should be as low as possible, while oxidative susceptibility (OS) as high as possible. In fact, in their work they compare Cox and OS values from different oils and, for example, Lycium chinense M. oil has Cox and OS values of 6.93 and 2983, respectively; Croatian Olive oil have values of 1.85 and 567; and Brazilian Soybean oil have Cox and OS values ranging from 6.41–6.94 and 2744–2987, respectively. Our results are in accordance with these studies.

  1. Line 256Authors stated that Cox values are low, however, if PI and OS values are high, Cox values should be high as well. What are the standard values that the authors referred to decide whether the values are high or low?

Previous studies have described these indices considering a range for low and high (Lycium chinense M. oil - Cox and OS values of 6.93 and 2983, respectively; Croatian Olive oil - values of 1.85 and 567; and Brazilian Soybean oil have Cox and OS values ranging from 6.41–6.94 and 2744–2987, respectively). We have followed these already published assumptions as a reference for the description of our results.

  1. Line 439: the sentence is not related to this study

This line was deleted.

  1. Line 24 and 26: V. vinifera cv. should be vinifera cv.

The authors acknowledge the reviewer for the careful revision. All “Vitis vinifera” are now in italic. 

Reviewer 3 Report

The authors have made an important effort to study vine leaves of different varieties. 

The authors should pay attention at same grammar errors, especially in the introduction. The literature on the phytochemical profile and the biological properties of vine leaves is very extensive and the authors should include more information and references in the introduction.

V. vinifera should be italics throughout the text. 

Why acetone was used for the extraction of pigments?

Please explain the use of atherogenic, thrombogenic, hypocholesterolemic/hypercholesterolemic and peroxidizability indexes because it is not at all clear.

Also the comparison with other plants could be deleted since it does not provide any significant conclusion.

The discussion is very indefinite, without presenting sufficiently the results. It describes mostly what should further studied.

Line 439, should be deleted.

Author Response

Reviewer 3:

The authors have made an important effort to study vine leaves of different varieties.

The authors should pay attention at same grammar errors, especially in the introduction. The literature on the phytochemical profile and the biological properties of vine leaves is very extensive and the authors should include more information and references in the introduction.

The authors acknowledge the reviewers’ comments. The manuscript was proofread.

The authors added more information and cited more literature in the introduction (Page 2, Line 52-75) as requested.

  1. vinifera should be italics throughout the text.

The authors acknowledge the reviewer for the careful revision. All “Vitis vinifera” are now in italic. 

Why acetone was used for the extraction of pigments?

The authors acknowledge the reviewers’ interest in the methodology used for pigment extraction. Acetone is one of the most widely used solvents for pigment extraction. An 80 to 100 percent acetone solution is a typical medium, since it allows stable results over 10 to 48 hours. The use of pure acetone causes the precipitation of chlorophyllase. This enzyme catalyzes the degradation of chlorophylls leading to the formation of chlorophyllide. Therefore, using pure acetone contributes to a decrease in the formation of this degradation product during the pigment extraction and to a more accurate measure of chlorophyll concentration (DOI: https://doi.org/10.1186/1746-4811-9-19).  Previous studies have already demonstrated that chlorophyllide formation could be minimized if leaf samples were ground in liquid nitrogen and extracted with acetone cooled to −20°C, which was the chosen method in the present work (DOI: https://doi.org/10.1016/j.febslet.2007.10.060). Regarding the extraction of carotenoids, it was previously shown that although acetone is slightly less efficient than tetrahydrofuran, it is the preferred solvent due to its low toxicity (https://doi.org/10.1007/s00449-019-02273-9).  

Please explain the use of atherogenic, thrombogenic, hypocholesterolemic/hypercholesterolemic and peroxidizability indexes because it is not at all clear.

The authors acknowledge the reviewers’ comments.

The objective of this work was to evaluate the nutritional potential of leaves from seven grapevine cultivars. Fatty acids are important nutrients in human physiology and possess a wide range of health promoting properties. These indexes (AI, TI, h/H and PI) are good indicators of lipid nutritional quality and stability. Hence, the authors calculated them in grapevine leaves to evaluate the lipid quality.

The reasons to analyse these indexes are stated in the discussion, in pages 10 and 11, lines 385-403, as follows:

“AI relates to the main classes of saturated (pro-atherogenic - favouring the adhesion of lipids to cells of the immunological and circulatory system) and unsaturated FAs (anti-atherogenic- inhibiting the aggregation of plaque and diminishing the levels of esterified FA, cholesterol, and phospholipids, thereby preventing the appearance of microcoronary and macrocoronary diseases). A low AI value ratio is recommended [26]. The TI index is the ratio between prothrombogenic (saturated) and antithrombogenic FA and relates to the tendency to form clots in the blood vessels. Both indexes indicate thea potential for stimulating platelet aggregation [27]. For human consumption, values below 1.0 (AI) and 0.5 (TI) are desirable [30]. AI and TI values in grapevine leaves are within the recommended values and similar to other green leafy vegetables such as broccoli (AI: 0.41; TI: 0.18), carrot (AI: 0.25; TI: 0.38) and asparagus (AI: 0.36; TI: 0.37).

Concerning the other indexes, nutritionally higher h/H values are considered more beneficial for human health.  The majority of the values found in grapevine leaves are higher than those found in macroalgae species, namely 1.26, 1.90, 2.09 and 4.22 for U. rigida, U. compressa, P. capillacea and G. microdon, respectively [44]. The OS, PI and Cox indexes are related to the susceptibility to oxidation of the FA. Grapevine leaves values for these indexes were according to the expected leading us to speculate that FA in grapevine leaves are not easily oxidated.”

Also the comparison with other plants could be deleted since it does not provide any significant conclusion.

The authors acknowledge the reviewers’ comments.

The main goal of our work was to improve the nutritional knowledge on this disregarded by-product, not only for the scientific community but also for the pharmaceutical and nutraceutical companies, winegrowers and consumers.

The vegetables used for the comparison are well established in the human everyday diet. Their nutritional properties are well known, and they are among the TOP health-promoting vegetables. The evaluation of the elemental, FA and pigment compositions and the comparison of the values obtained in our work with other vegetables is important as it clearly indicates the nutritional potential of grapevine leaves. Since the values obtained for grapevine leaves in our study are similar to these vegetables, the consumption of grapevine leaves can also be associated with several positive effects on human health.

The discussion is very indefinite, without presenting sufficiently the results. It describes mostly what should further studied.

The authors acknowledge the reviewers’ comments.

Up to our knowledge this it is the first study that analyse the leaves of a diverse group of genotypes (differing on berry colour and country of origin) and measure their element and pigment compositions. Since our work is a preliminary study, and our main goal is to valorise grapevine leaves for human consumption, pharmaceutical and cosmetic industries, the comparison of our results with already described in the literature for other vegetables is important for this valorisation. Moreover, the authors consider that further studies regarding, for instance, a larger set of cultivars and different grapevine developmental stages are essential to completely unveil the full capacity of grapevine leaves as nutritional assets. Therefore, the authors consider that it is highly important to indicate the next steps to be taken to fully understand the value of grapevine leaves.

Line 439, should be deleted.

This line was deleted.

Round 2

Reviewer 2 Report

Refer to the answers and changes given by the authors in responding to point 3 in the first review, I do not agree the statements in line 258-260, 274-278, and 401-403: “While Cox values should be low, indicating that FAs are less prone to oxidation, OS should be as high as possible”; “Regarding lipid oxidation, FAs present in grapevine leaves are stable and do not oxidize easily as OS and PI values in grapevine leaves are high, above 5000 and 100 in all genotypes, respectively. In addition, Cox values are low and consistent in all V. vinifera cultivars. In fact, higher values of OS and PI as well as lower values of Cox represent less lipid susceptibility to oxidation”; “Grapevine leaves values for these indexes were in accordance with the expected, leading us to speculate that FA in grapevine leaves are not easily oxidized

According to the formulas for Cox, OS and PI calculation (formula 3,4,6), the ratio multiply by the fatty acid follows the number of double bond in it. The more the double bond, the higher ratio multiply (ie. ?? = ???? + 45 × ?18: 2 + 100 × ?18: 3: the ratio multiply by MUFA is 1; the ratio multiply by ?18: 2 is 45, and the ratio multiply by ?18: 3 is 100). Due to the fact that the more double bond in the fatty acid, the easier fatty acid being oxidized, therefore, the ratio multiply by C18: 3 in the formula is greater. This indicate that the higher Cox, OS, PI values, the greater the oxidation happen to the fatty acid.

These concepts are also supported by “Factors Affecting Neurological Aging: Genetics, Neurology, Behavior, and Diet” by Colin R. Martin, Victor R. Preddy, and Rajkumar Rajendram, “Peroxidizability index is a measure of the relative susceptibility of a given membrane fatty acid composition to peroxidative damage. The higher the peroxidizability index value, the greater the susceptibility to lipid peroxidation.” In another book “Oxidative Stress and Digestive Diseases” by Toshikazu Yoshikawa, “The more highly unsaturated the fatty acid is, the greater is the peroxidizability.” The book “Food Lipids: Chemistry, Nutrition, and Biotechnology, Third Edition” by David B. Min, also mentioned that “the susceptibility of PUFA to oxidation depends on the availability of bis-allylic hydrogens, oxidative stability of each PUFA is inversely proportional to the number of bis-allylic positions in the molecule or the degree of unsaturation of the PUFA.”

Author Response

Answer to Reviewer 2:

Refer to the answers and changes given by the authors in responding to point 3 in the first review, I do not agree the statements in line 258-260, 274-278, and 401-403: “While Cox values should be low, indicating that FAs are less prone to oxidation, OS should be as high as possible”; “Regarding lipid oxidation, FAs present in grapevine leaves are stable and do not oxidize easily as OS and PI values in grapevine leaves are high, above 5000 and 100 in all genotypes, respectively. In addition, Cox values are low and consistent in all V. vinifera cultivars. In fact, higher values of OS and PI as well as lower values of Cox represent less lipid susceptibility to oxidation”; “Grapevine leaves values for these indexes were in accordance with the expected, leading us to speculate that FA in grapevine leaves are not easily oxidized”

According to the formulas for Cox, OS and PI calculation (formula 3,4,6), the ratio multiply by the fatty acid follows the number of double bond in it. The more the double bond, the higher ratio multiply (ie. ?? = ???? + 45 × ?18: 2 + 100 × ?18: 3: the ratio multiply by MUFA is 1; the ratio multiply by ?18: 2 is 45, and the ratio multiply by ?18: 3 is 100). Due to the fact that the more double bond in the fatty acid, the easier fatty acid being oxidized, therefore, the ratio multiply by C18: 3 in the formula is greater. This indicate that the higher Cox, OS, PI values, the greater the oxidation happen to the fatty acid.

These concepts are also supported by “Factors Affecting Neurological Aging: Genetics, Neurology, Behavior, and Diet” by Colin R. Martin, Victor R. Preddy, and Rajkumar Rajendram, “Peroxidizability index is a measure of the relative susceptibility of a given membrane fatty acid composition to peroxidative damage. The higher the peroxidizability index value, the greater the susceptibility to lipid peroxidation.” In another book “Oxidative Stress and Digestive Diseases” by Toshikazu Yoshikawa, “The more highly unsaturated the fatty acid is, the greater is the peroxidizability.” The book “Food Lipids: Chemistry, Nutrition, and Biotechnology, Third Edition” by David B. Min, also mentioned that “the susceptibility of PUFA to oxidation depends on the availability of bis-allylic hydrogens, oxidative stability of each PUFA is inversely proportional to the number of bis-allylic positions in the molecule or the degree of unsaturation of the PUFA.”

The authors acknowledge the reviewer’s comments and thorough revision.

As the sentences referred may lead to misinterpretations by the readers, they were changed as follows:

Page 8, Lines 324-326: Regarding lipid oxidation, OS values in grapevine leaves were high, above 5000, in all genotypes and Cox values were low and consistent in all V. vinifera cultivars. The PI value was around 100 in all genotypes.     

Page 11, Lines 478-495: Grapevine leaves values for OS and Cox indexes were in accordance with the expected. In E. Covaci et al. (DOI:10.3233/JBR-190450), the authors refer that the oxidisability value (Cox) should be as low as possible, while oxidative susceptibility (OS) as high as possible. In fact, in their work they compare Cox and OS values from different oils and, for example, Lycium chinense M. oil has Cox and OS values of 6.93 and 2983, respectively; Croatian Olive oil have values of 1.85 and 567; and Brazilian Soybean oil have Cox and OS values ranging from 6.41–6.94 and 2744–2987, respectively. Our results are in accordance with these studies, leading us to speculate that FA in grapevine leaves have a high oxidative protective potential. The PI value in grapevine leaves is similar to the ones found in different everyday dietary oils. For instance, in soybean, perilla and fish oils have PI values of 70.78, 134.82 and 262.08, respectively. Although it is described in the literature that that higher values of PI, the greater the susceptibility to lipid peroxidation, Min Jeong Kang and co-workers proved that a high polyunsaturated/saturated fatty acid ratio diet has a beneficial effect on cardiovascular disease risk even without antioxidant when the PI value is the same, above 80 (https://doi.org/10.1016/j.cccn.2004.07.005). This is because a high P/S ratio diet makes it difficult to increase lipid peroxidation because of the high concentrations of PUFA (https://doi.org/10.1002/biof.5520130104). There are very few studies about what the optimal PI value might be, hence further studies are encouraged to evaluate this index in the everyday die

Reviewer 3 Report

The authors have followed the recommendations and answered all the questions. The manuscript can be accepted in this format.

Author Response

The authors acknowledge reviewer comments and contribution for the improvement of our manuscript.